# Comparison of Point-of-Care Ultrasonography and Radiography in the Diagnosis of Long-Bone Fractures

**DOI:** 10.3390/medicina55070355

**Published:** 2019-07-09

**Authors:** Mustafa Avci, Nalan Kozaci, Gul Tulubas, Gulsum Caliskan, Aysegul Yuksel, Adeviye Karaca, Fatih Doganay, Ibrahim Etli

**Affiliations:** 1Department of Emergency Medicine, Antalya Education and Research Hospital, University of Health Sciences, Antalya 07100, Turkey; 2Department of Emergency Medicine, Ercis State Hospital, Van 65400, Turkey; 3Department of Emergency Medicine, Edremit State Hospital, Balikesir 10300, Turkey; 4Department of Orthopedics and Traumatology, Antalya Education and Research Hospital, University of Health Sciences, Antalya 07100, Turkey

**Keywords:** bone ultrasonography, diagnosis of fracture with ultrasonography, fracture characteristics, long-bone, point-of-care ultrasonography, POCUS, trauma ultrasonography

## Abstract

*Background and objectives:* In this study, the accuracy of point-of-care ultrasonography (POCUS) was compared to radiography (XR) in the diagnosis of fractures, the determination of characteristics of the fractures, and treatment selection of fractures in patients admitted to the emergency department (ED) due to trauma and suspected long bone (LB) fractures. *Materials and Methods:* The patients were included in the study, who were admitted to ED due to trauma, and had physical examination findings suggesting the presence of fractures in LB (humerus, radius, ulna, femur, tibia, and fibula). The patients were evaluated by two emergency physicians (EP) in ED. The first EP examined LBs with POCUS and the second EP examined them with XR. LBs were evaluated on the anterior, posterior, medial, and lateral surfaces and from the proximal joint to the distal one (shoulder, elbow, wrist, hip, knee, and ankle joint) in both longitudinal and transverse axes with POCUS. *Results:* A total of 205 patients with suspected LB fractures were included in the study. LB fractures were determined in 99 patients with XR and in 105 patients with POCUS. The sensitivity, specificity, positive predictive value, negative predictive value of POCUS in determining the fractures were 99%, 93%, 93%, and 99%, respectively, compared to XR. Compared to XR, POCUS was able to determine 100% of fissure type fractures (kappa (κ) value: 0.765), 83% of linear fractures (κ: 0.848), 92% of fragmented fractures(κ: 0.756), 67% of spiral fractures (κ:0.798), 75% of avulsion type fractures (κ: 0.855), and 100% of full separation type fractures (κ: 0.855). *Conclusions:* This study has demonstrated that POCUS has a high sensitivity in diagnosing LB fractures. POCUS has a high sensitivity in identifying fracture characteristics. POCUS can be used as an alternative imaging method to XR in the diagnosis of LB fractures and in the determination of fracture characteristics.

## 1. Introduction

Orthopedic injuries in all ages are common causes of admission in emergency departments (ED), and long-bone (LB) injuries are frequently encountered [1,2,3]. Radiography (XR) is the first-line diagnostic imaging technique in the ED to evaluate LB fractures. The characteristics of a fracture are usually identified in the posterior-anterior, lateral, and oblique XR images. When XR images fail to offer adequate information, computed tomography (CT) can be used [4]. However, CT can be time-consuming in ED, where the patient volume is high, and it may extend the patient waiting time. Furthermore, exposure to ionizing radiation is of concern, especially in special groups of individuals including children and pregnant women.

Ultrasonography (US) is currently used in ED for the management of trauma patients and in the resuscitation processes, as well as for identifying abdominal, cardiothoracic, and vascular injuries [5,6]. US is advantageous because it is easily accessible, inexpensive, reproducible, portable, free of ionizing radiation, and it provides real-time imaging.

The point-of-care ultrasonography (POCUS) has recently been introduced to be utilized for evaluation of the musculoskeletal tissue. It has been demonstrated that US provides significant advantages especially in the pediatric population susceptible to radiation, at the pre-hospital settings, and in pregnant women, as well as reducing exposure to serial XR in the process of fracture reduction [7]. Furthermore, POCUS is not only used for imaging bone injuries but it is also utilized for visualizing ligaments, tendons, and soft tissues [5]. Studies comparing POCUS to XR and CT have demonstrated that POCUS is highly sensitive in diagnosing fractures and identifying fracture characteristics [1,4,5,7,8]. In the decision-making process for the treatment of LB fractures, along with the determination of the fracture, it is important to identify the type of the fracture, its location, whether the fracture extends to the joint space, any presence of accompanying adjacent bone fractures, angulation, stepping-off, and soft tissue injuries. However, there are no studies in the literature investigating the presence of a fracture and identifying its characteristics in patients with LB trauma.

In this study, the diagnostic accuracy of POCUS is compared to XR in the diagnosis of fractures, the determination of characteristics of the fractures, and the treatment selection of fractures in patients admitted to ED due to trauma and suspected LB fractures.

## 2. Materials and Methods

This is a study of a diagnostic test comparison, which was conducted in ED of a tertiary hospital in the period between January 2018 and April 2019 after obtaining the ethics committee approval. The study was approved by the University of Health Sciences Antalya Education and Research Hospital Clinical Research Ethics Committee (Antalya, Turkey) with the registry number 2/2 on 25 January 2018. The patients in all ages were included in the study, who were admitted to ED due to trauma, had stable vital signs and physical examination findings suggesting the presence of fractures in LB (humerus, radius, ulna, femur, tibia, and fibula). A written informed consent form was obtained from all study patients or their relatives. The exclusion criteria were (a) having XR at external healthcare centers prior to admission to our hospital, (b) the presence of open fractures, (c) the presence of neurovascular injuries, (d) the presence of dislocations with the fracture, (e) having unstable vital signs or any other life-threatening injuries, (f) being pregnant, (g) having severe pain during the POCUS examination, and (h) not consenting to participate in the study.

Prior to the commencement of the study,5 emergency physicians (EP), who participated the study as investigators and who had an ED working experience of at least 3 years, received one hour of theoretical and one hour of practical training about the examination and evaluation of LBs in XR images. Then, these EPs, who used already ultrasound in the patient management in the ED, received one hour of theoretical and one hour of practical training about the examination and evaluation of LBs with POCUS. The practical training addressed both intact and fractured bones. Furthermore, the EPs, who would perform the POCUS examinations, performed test examinations during the three weeks prior to the commencement of the study.

The patients were included in the study during the days when the EPs who participated in the study worked in the ED trauma room. The patients were evaluated by two EPs in ED. The first EP examined LBs with POCUS and the second EP examined them with XR.

A standard data record form was created for the study. The physical examination findings (point of tenderness, edema, ecchymosis, crepitation, deformities, and abnormal range of motion or neurovascular damage) of the patients were evaluated and recorded by two EPs. Then, the first EP evaluated the LBs with POCUS based on the physical examination findings. The 7.5 MHz linear probe of a standard ultrasound device (Mindray DC-T6) was used in the evaluation of the LBs with POCUS. LBs were evaluated on the anterior, posterior, medial, and lateral surfaces and from the proximal joint to the distal one (shoulder, elbow, wrist, hip, knee, and ankle joint) in both longitudinal and transverse planes with POCUS. POCUS examination of the LBs was performed in eight steps, according to the “Kozaci Protocol” (Table 1).

The angulation and the stepping-off distance of the fractures were measured using the standard software of the US device. The angulation was determined based on the angle formed by two lines, drawn along the cortical edges of the fracture surfaces. The stepping-off distance of the fracture was recorded by measuring the distance between the fractured and the healthy cortices. Repeated examinations were performed in the areas of point of tenderness during the POCUS examination. The findings were confirmed by making comparisons with the intact extremity. Each of these steps took approximately 2 min.

The XR images of the patients were interpreted by the second EP. XR images of the LBs were evaluated in eight steps according to the “Modified Kozaci Protocol” (Table 2). The interpretations made to the XR images by second EP were controlled and verified by a clinical orthopedics and traumatologist. The two EPs, who evaluated the LBs with POCUS and XR, were double-blinded to each other’s interpretations.

The EP, who interpreted the XR images of the patients, decided the patients’ ED management and the final treatment. The study protocol neither intervened on any diagnostic or treatment procedures provided to the patients nor posed any risks to them.

### Statistical Analysis

A total of 216 forms were fulfilled in for the purposes of the study. The POCUS examination could not be completed in 5 patients due to extremely severe pain. Three patients who had their XR images taken at external healthcare centers, two patients who had unstable vital signs, and one patient who was pregnant were excluded from the study (Figure 1). A total of 205 patients were included in the statistical analysis.

Analysis of the data collected in the study was performed using Package for the Social Sciences 21 statistical software package (IBM Corporation, Chicago, IL, USA). The sensitivity, specificity, positive predictive value (PPV), negative predictive value (NPV), and the Kappa (κ) coefficient of POCUS were calculated comparing to XR. The ROC curve was obtained by ROC analysis (the state variable is XR and the dependent (test) variable is POCUS). Concordance was graded according to the κ coefficient. A κ value of greater than 0.75 was considered as perfect concordance, 0.75–0.40 as moderate concordance, and less than 0.40 as poor concordance. In order to determine the statistical significance, *p* < 0.05 with 95% confidence intervals was considered significant in the analysis. For descriptive statistics, data obtained using the Chi-square test (X^2^) and kappa statistics were compared.

## 3. Results

A total of 205 patients with suspected LB fractures were included in the study. Of the study patients, 98 (49%) were females and 107 (52%) were males. The mean age of the patients was 33 ± 22 years. Of the patients, 69 (34%) were 18 years old or younger. The most common physical examination finding in the patients was the combination of edema and point of tenderness (Table 3).

LB fractures were determined in 99 patients with XR and in 105 patients with POCUS. The LB fractures were determined to be the most common in the radius and the second most common fracture was in the humerus (Table 4). Concurrent radius-ulna and tibia-fibula fractures were determined in XR images in 9 patients and in 1 patient, respectively. In addition, fractures were determined with XR in the patella of 2 patients, in the clavicle of 1 patient, and in the calcaneus bone of 1 patient.

The sensitivity, specificity, PPV, and NPV of POCUS in determining the fractures were 99%, 93%, 93%, and 99%, respectively, compared to XR (*p* < 0.001, κ: 0.922, AUC: 0. 962, 95%, and CI: 0.932–0.992). The most common fracture type was linear fractures (Table 5, Figure 2, and Appendix A). Compared to XR, POCUS was able to determine 100% of fissure type fractures (X^2^, *p* < 0.001, and κ: 0.765), 83% of linear fractures (X^2^, *p* < 0.001, and κ: 0.848), 92% of fragmented fractures(X^2^, *p* < 0.001, and κ: 0.756), 67% of spiral fractures (X^2^, *p* < 0.001, and κ: 0.798), 75% of avulsion type fractures (X^2^, *p* < 0.001, and κ: 0.855), and 100% of full separation type fractures(X^2^, *p* < 0.001, and κ: 0.855) (Table 6). According to the κ value, a perfect concordance between POCUS and XR in identifying all types of fractures was determined.

In the POCUS examination, 81 (40%) patients had soft tissue edema and hematoma. Hemarthrosis was determined in 1 patient with a proximal tibial fracture and in 2 patients with distal humerus fractures. Pain and point of tenderness were determined in 118 (58%) patients during the POCUS examinations.

LB fractures were mostly determined at a distal localization (Table 7). The XR images revealed that the fractures extended to the joint space in 15 (7%) patients, the fractures involved the epiphyseal line in 7 (3%) patients, the fractures were angulated in 45 (22%) patients, there was a stepping-off distance of the fracture in 39 (19%) patients, and there were accompanying adjacent bone fractures in 14 (7%) patients. The sensitivity of POCUS was found to be low in determining the fractures extending to the joint space (Table 8). According to the κ value, a perfect concordance between POCUS and XR in angulation (κ: 0.871), stepping off (κ: 0.953), and involvement of the epiphyseal line (κ: 0.921) were determined. This concordance was moderate in the extension of the fracture into the joint space (κ: 0.643).

Of the patients, 31 (15%) were hospitalized to undergo surgical interventions. The rest of the study patients were discharged from the ED after the reduction and splint placement in 24 (12%) patients, only splint placement in 97 (47%) patients, and elastic bandage placement in 53 (26%) patients.

## 4. Discussion

Delays in the diagnosis and treatment of LB fractures may lead to organ loss and even death [2,3]. A thorough physical examination including a neurovascular examination should be performed to make an early and accurate diagnosis. One of the major advantages of POCUS is that it is possible to communicate with the patient and it allows us to perform a dynamic examination. During POCUS, it is possible to determine the areas where point of tenderness emerges upon ultrasound examination, which provides the opportunity to re-evaluate these areas. Furthermore, POCUS can show hemarthrosis in joint spaces and hematoma in traumatic tissue. One of the major disadvantages of POCUS is that point of tenderness may emerge during the US examination and it may turn into pain consequently. Severe pain can hinder performing a POCUS examination. In our study, edema and point of tenderness were the most common findings in the physical examination. However, the determination of the fractures was enhanced since the number of findings in the physical examination increased. The POCUS examination performed after the physical examination revealed hemarthrosis in 1 patient with a proximal tibia fracture and in 2 patients with distal humerus fractures. Determination of hemarthrosis is critical because it can change the treatment decision. In our study, the POCUS examination revealed pain and point of tenderness in 58% of the patients. However, the POCUS examination could not be completed due to severe pain only in 5 patients.

The US imaging shows the bone fractures as an interruption in the cortical continuity or an irregularity in the cortex in longitudinal scanning. Furthermore, impaired cortical continuity is evident in transverse screening. Several studies have found out that US is able to determine cortical deformations more easily compared to XR, and that its sensitivity and specificity are high in determining fractures [4,8,10]. A study, comparing the accuracy of US to XR in determining LB fractures, found the sensitivity and specificity of US as 73% and 92%, respectively [10]. That study reported that the fractures occurred most commonly in the radius and fibula. Another study reported the sensitivity, specificity, PPV, and NPV of POCUS as 100%, 93%, 88%, and 100%, respectively, in comparison to XR in order to determine tibia fractures. Sensitivity, specificity, PPV, and NPV of POCUS in determining a fibula fracture were 100%, 97%, 96%, and 100%, respectively [8]. A similar study found that the sensitivity of POCUS in determining distal radius fractures was 98%, specificity was 96%, PPV was 98%, NPV was 96%, and accuracy of the test was 98% [4]. Similar to other studies, the sensitivity, specificity, PPV, and NPV of POCUS were found as 99%, 93%, 93%, and 99%, respectively, in determining the LB fractures in our study.

The selected treatment approach varies based on the type of the fracture whether it is a torus, greenstick, fissure, linear, fragmented, spiral, or full separated type fracture. In children, LB fractures tend to occur as incomplete fractures, called greenstick fractures, where one side of the bone is fractured and the other side is simply bent. In contrast, adult fractures typically tend to result from a blast effect, usually generated by a direct and high-energy impact on the bone and the surrounding soft tissues. When the fracture traverses a bone from one end to the other, it is called a “linear fracture,” and, when the bone is fractured and separated into several parts, it is called a “fragmented fracture.” Bending traumas can cause “spiral fractures” and a single, large, free-floating bone segment between two well-defined fracture lines is called a “segmental fracture” [11]. In our study, the most common fracture type in the patients was the linear fracture. Compared to XR, POCUS was able to determine 100% of the fissure type fractures, 83% of the linear fractures, 92% of the fragmented fractures, 67% of the spiral fractures, 75% of the avulsion type fractures, and 100% of the full separation type fractures. According to the κ value, a perfect concordance between POCUS and XR in identifying all types of fracture was determined. The reason for failure in determining fractures with POCUS may be that the patients move their limbs due to pain during the examination. As a matter of fact, 58% of the patients stated that they felt pain during the POCUS examination. In addition, the POCUS examination could not be completed due to severe pain in only 5 patients.

The location of the fracture is a critical factor in deciding the mode of treatment for LB fractures. The fractures are called according to their location as proximal, distal, and shaft fractures. A fracture, located just above the distal humerus or above the femoral condyles, is called a supracondylar fracture. Other bone markers include the head of the radius in the elbow, the radial styloid process in the wrist, and the greater tuberosity of the humerus. It is possible to give numerous additional examples [11]. In our study, most of the fractures were determined at distal locations on the bones and there was no confusion in identifying the proximal, distal, and shaft locations.

Concurrent dislocations or adjacent bone fractures may accompany LB fractures. There may be fractures involving both the distal pole of one bone and the proximal pole of the adjacent bone. Therefore, the adjacent bone should also be examined when the location of the fracture is evaluated. In our study, 7% of the patients had an accompanying adjacent bone fracture. Among the accompanying adjacent bone fractures,9 patients had distal ulna fractures and distal radius fractures, 1 patient had a distal fibula fracture and a distal tibia fracture. In addition, XR images determined fractures in the patella of 2 patients, in the clavicle of 1 patient, and in the calcaneus bone of 1 patient.

The treatment decision varies based on several factors including the location of the fracture, the type of fracture (fissure, linear, fragmented, avulsion, etc.), the extension of the fracture to the joint space, whether the fracture involves the epiphyseal line, angulation of the fracture, the stepping-off distance of the fracture, and the presence of a dislocation with the fracture. For example, treatment includes a closed reduction with splint/cast immobilization for radius fractures if there is an extra-articular radial shortening of <5 mm, and a dorsal angulation of <5° or a dorsal angulation within 20° of the contralateral distal radius. Conversely, surgery is recommended in the presence of the following: a dorsal angulation of >5mm, a contralateral distal radius volar angulation of >20°, intra-articular fractures of >2mm, radial shortening of >5mm, accompanying adjacent bone fractures in the ulna, and accompanying fragmented and displaced extra-articular fractures (Smith’s fractures, etc.) [3,4,8,11,12]. Therefore, identifying the characteristics of a fracture with POCUS in addition to determining the fracture, is critical in the decision process for treatment. POCUS has been demonstrated to be successful in determining the characteristics of bone fractures [1,4,5,8]. A study, investigating the metatarsal fractures, found the sensitivity of POCUS as 100% in determining the presence of angulation and 83% in determining the stepping-off distance of the fracture [7]. In a study comparing POCUS and CT in elbow injuries, sensitivity, specificity, PPV, and NPV of POCUS in determining the extension of the fracture to the joint space or involvement of the epiphyseal line were 100%, 97%, 92%, and 100%, respectively [1].In our study, the sensitivity, specificity, PPV, and NPV of POCUS in determining angulation were 89%, 98%, 91%, and 97%, respectively. The sensitivity, specificity, PPV, and NPV of POCUS in determining stepping-off were 97%, 99%, 95%, and 99%, respectively. The sensitivity, specificity, PPV, and NPV of POCUS were found to be 60%, 98%, 75%, and 97%, respectively, in determining the extension of the fracture to the joint space. The sensitivity, specificity, PPV, and NPV of POCUS were 86%, 100%, 100%, and 99%, respectively, when determining the involvement of the epiphyseal line. The κ value was 0.921 in a perfect concordance for determining the involvement of the epiphyseal line. In contrast, there was a moderate concordance between POCUS and XR, the κ value was identified to be 0.643 in the extension of the fracture to the joint space. These results suggest that POCUS is insufficient in determining the extension of the fracture to the joint space.

The borders of a compartment are often made up of bones or tissues. The capacities of the compartments against stretch are minimal. Acute compartment syndrome (ACS) is a state of raised pressure in the muscle compartments of the extremities. In ACS, the mechanisms responsible for the compartment pressure increasing are tissue edema and hematoma. Numerous studies have shown that fractures, soft tissue injuries, and crush syndrome are highly common causes of ACS. Therefore, ACS should be considered in LB fractures. ACS occurs most commonly in the lower extremities in the region where the tibia and fibula are present [4,13]. For the reasons listed above, EP should examine whether ACS is present while performing a physical examination in patients with suspected bone fractures. Unlike XR, the ability to examine the soft tissue is the advantage of US. In our study, 81 (40%) patients had soft tissue edema and hematoma identified in the POCUS examination. These patients were considered to be likely to develop ACS, and, therefore, they were followed-up in this respect.

In our study, it was demonstrated that the POCUS can be used to diagnose LB fractures with short-term training. In other studies similar to our study, it was demonstrated that the POCUS had high accuracy rates with short-term trainings for diagnosing bone fractures [6,7,9]. In addition, in a study comparing POCUS to CT in determining elbow injuries, POCUS had a high accuracy rate after short-term training [1].Therefore, POCUS may be an alternative imaging method to XR and may reduce the requirement of CT. Especially in selected patients, POCUS can be performed before CT view.

## 5. Conclusions

This study demonstrated that POCUS has high sensitivity and specificity in diagnosing LB fractures. Therefore, POCUS can be used as an alternative or complementary imaging method to XR in determining LB fractures. However, the accuracy rate of POCUS is low especially in identifying the extension of the fracture to the joint space. Further studies are needed.

### Limitations

Examining the epiphyseal line in detail is important when analyzing the fracture characteristics. In our study, fractures involving the epiphyseal line could not be analyzed extensively because the number of patients aged 18-years-old or younger was relatively low. In addition, the number of patients who had fractures extending to the joint space is low. More valid results can be given with larger studies.

## Figures and Tables

**Figure 1 medicina-55-00355-f001:**
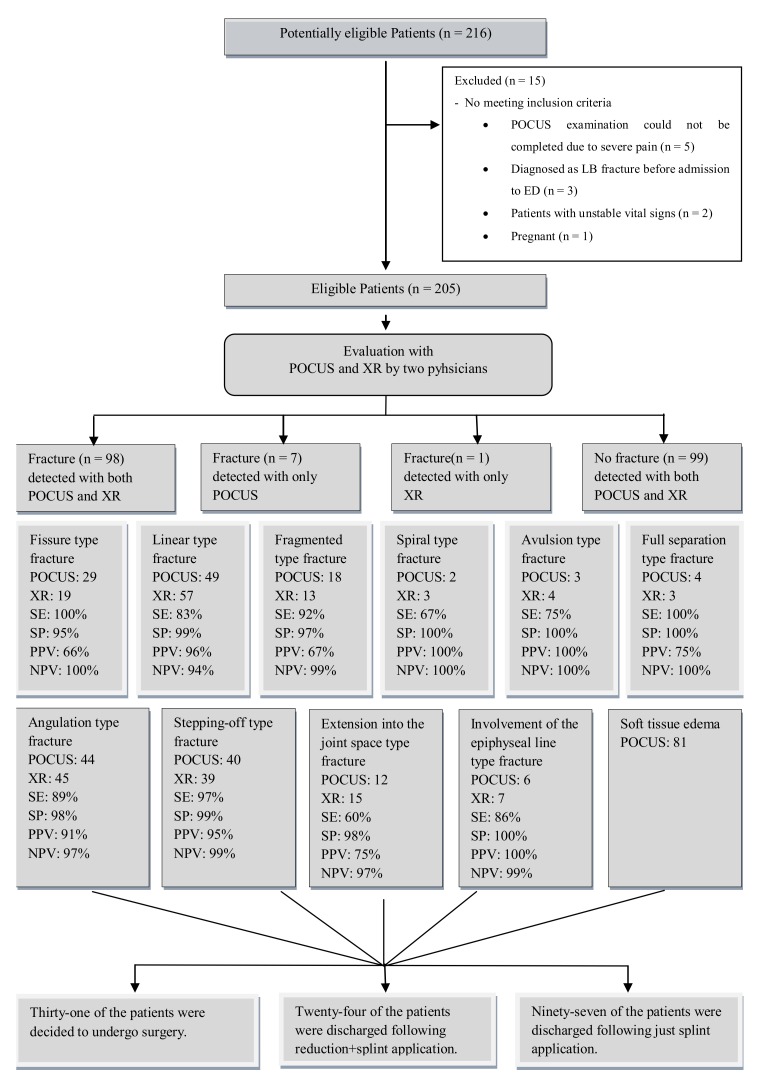
Study flow diagram. SE: sensitivity; SP: specificity; PPV: positive predictive value; NPV: negative predictive value.

**Figure 2 medicina-55-00355-f002:**
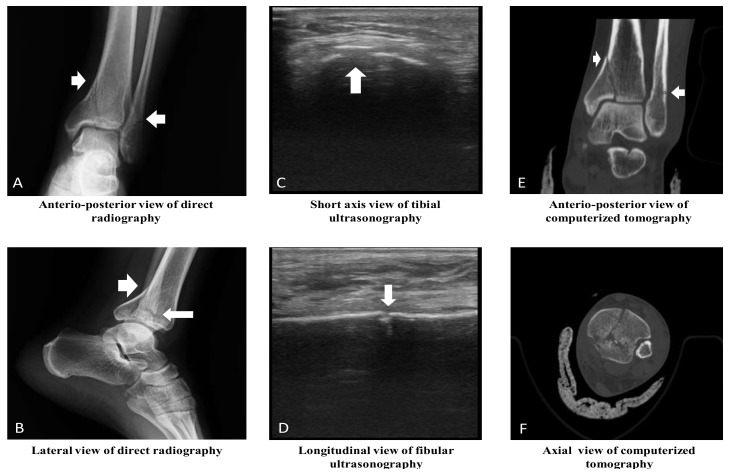
Radiological images of ankle joint in a 33-year-old male patient. (**A**) Tibial linear fracture with extension into the joint space and fibular linear fracture. (**B**) Tibial linear fracture with stepping-off and angulation, and fibular linear fracture. (**C**) Tibial fragmented fracture with stepping-off and angulation. (**D**) Fibular linear fracture. (**E**) Tibial linear fracture with extension into the joint space, and fibular linear fracture. (**F**) Tibial fragmented fracture, and fibular linear fracture. (Note: The clinical orthopedics and traumatologist thought that computed tomography is required for determining the characteristic of fractures and treatment decision of fractured bones.).

**Table 1 medicina-55-00355-t001:** Kozaci protocol for determination of fractures with POCUS [8].

Kozaci Protocol for Determination of Fractures with POCUS
1. Determining the presence of fractures (Cortical disruption)
2. Determining the type of fracture (fissure, linear, fragmented, spiral) and localization
3. The angulation of the fracture
4. The stepping-off distance of the fracture
5. The extent of the fracture to the joint space
6. Control of the fracture if it contains the epiphyseal line or not
7. Control of accompanying adjacent bone fracture
8. Control of the presence of hematoma in the soft tissue and joint space

POCUS: Point-of-care ultrasound.

**Table 2 medicina-55-00355-t002:** Modified Kozaci protocol for determination of fractures with XR [9].

Modified Kozaci Protocol for Determination of Fractures with XR
1. Determining the presence of fractures (Cortical disruption)
2. Determining the type of fracture (fissure, linear, fragmented, spiral) and localization
3. The angulation of the fracture
4. The stepping-off distance of the fracture
5. The extent of the fracture to the joint space
6. Control of the fracture if it contains the epiphyseal line or not
7. Control of the accompanying adjacent bone fracture
8. Control of the joint space and the presence of joint dislocation

XR: Radiography.

**Table 3 medicina-55-00355-t003:** Comparison of the physical examination findings of the patients with the presence of fractures that were determined by POCUS and XR.

Physical Examination Findings	Fractures Determined by POCUS, *n* (%)	Fractures Determined by XR, *n* (%)	Total
Neurovascular injury	-	-	0
Point of tenderness	16 (7.8)	15 (7.3)	54
Edema + point of tenderness	48 (23.4)	43 (21.0)	105
Deformity + edema + point of tenderness	24 (11.7)	24 (11.7)	28
Crepitations +deformity + edema + point of tenderness	17 (8.3)	17 (8.3)	18
Total	105 (51.2)	99 (48.3)	205

POCUS: Point-of-care ultrasound, XR: Radiography.

**Table 4 medicina-55-00355-t004:** The bones were a determined fracture by POCUS and XR.

Bones	POCUS	XR
Humerus	14	14
Radius	48	44
Ulna	4	4
Radius + ulna	9	9
Femur	8	8
Tibia	8	9
Fibula	11	10
Tibia + fibula	2	1
Total	105	99

POCUS: Point-of-care ultrasound, XR: Radiography.

**Table 5 medicina-55-00355-t005:** Long bone fracture type according to POCUS and XR.

POCUS (*n* = 105)	**Fracture Type**	**XR (*n* = 99)**						
		**No Fracture**	**Fissure**	**Linear**	**Fragmented**	**Spiral**	**Avulsion**	**Full Separation**
	No fracture	99	-	-	-	1	-	-
	Fissure	6	19	3	-	-	1	-
	Linear	1	-	47	1	-	-	-
	Fragmented	-	-	6	12	-	-	-
	Spiral	-	-	-	-	2	-	-
	Avulsion	-	-	-	-	-	3	-
	Full separation	-	-	1	-	-	-	3

POCUS: Point-of-care ultrasonography, XR: Radiography.

**Table 6 medicina-55-00355-t006:** Diagnostic accuracy of POCUS in determining the fracture type.

Fracture Type	Sensitivity (%)	Specificity (%)	PPV	NPV	AUC	95% CI
Fissure	100	95	66	100	0.973	0.953–0.993
Linear	83	99	96	94	0.906	0.846–0.965
Fragmented	92	97	67	99	0.946	0.861–1.000
Spiral	67	100	100	100	0.833	0.506–1.000
Avulsion	75	100	100	100	0.875	0.619–1.000
Full separation	100	100	75	100	0.998	0.991–1.000

POCUS: point-of-care ultrasound. PPV: positive predictive value. NPV: negative predictive value. AUC: area under the curve. CI: confidence interval.

**Table 7 medicina-55-00355-t007:** Localization of long bone fractures.

Imaging Technique	Proximal, *n* (%)	Shaft, *n* (%)	Distal, *n* (%)	Total, *n* (%)
POCUS	22(21)	11 (10)	72 (69)	105
XR	19 (19)	12 (12)	68 (69)	99

POCUS: point-of-care ultrasound, XR: radiography.

**Table 8 medicina-55-00355-t008:** Diagnostic accuracy of POCUS in determining fracture characteristics.

Fracture Characteristics	Sensitivity (%)	Specificity (%)	PPV	NPV	AUC	95% Cl	*p*
The extension of the fracture to the joint space	60	98	75	97	0.792	0.637–0.947	<0.001
Involvement of the epiphyseal line	86	100	100	99	0.929	0.776–1.000	<0.001
Angulation	89	98	91	97	0.932	0.876–0.987	<0.001
Stepping-off	97	99	95	99	0.981	0.951–1.000	<0.001
Adjacent bone fracture	93	99	87	99	0.959	0.880–1.000	<0.001

POCUS: point-of-care ultrasound, PPV: positive predictive value, NPV: negative predictive value, AUC: area under the curve, CI: confidence interval.

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
