# Peer review of "Comparison of Point-of-Care Ultrasonography and Radiography in the Diagnosis of Long-Bone Fractures"

_1010-660X, 2019, doi:10.3390/medicina55070355_

Reviewer 1 Report

this is a well written study comparing POCUS to X ray exams for fracture detection in the Emergency Dpt

In the Results section it is suggested aPart from CI to include a p value as well which may be more familiar to the readers

 In the discussion section authors need to make a comment upon the potential utility of POCUS as detection only exam that will send selected and diagnosed patients to further examination by CT. The reason for that is that sensitivity and specificity of the technique are  decreased when fragmented or spiral fractures are concerned and orthopedic surgeons will require ipmortant info obtained by multiplanar CT views prior to surgical reduction

Include a limitation paragraph

Reviewer 2 Report

This is a paper on the use of US in the diagnosis of fracture.

Some points:

line 62: "efficacy": I think "diagnostic accuracy" should be better

line 66: I think that this is a study of diagnostic test comparison.

line 114. in the statistic you do not mention ROC curve.

line 125: Chi-square and K statistic are not shown in the results.

line 142: positive predicted value (PPV) and negative predicted value (NPV) should be reported together with sensitivity (SE) and specificity (SP).

line 144-145: the percentages are not enougth to show the diagnostic accuracy of POCUS Vs Rx for the fracture types reported in those lines. SE, SP, PPV, NPV should be also reported.

line 215: "POCUS was able to determine 100 % of fissure fracture". It is true but misleading. In fact, POCUS ranks 29 fissures on 19 classified by Rx. This means that only 65.5% of the fractures classified by POCUS correspond to those detected by RX (PPV). From a clinical point of view this is important to be considered. 34.5% of fracture detected by POCUS are false positives!  Thus the clinician is not at all sure that the fracture detected by POCUS is actually fissure type also at  RX.

Similar considerations apply to the other fractures of lines 216 and 217

Figure 1. the number of fractures of line 5 does not match with that of line 6.

Table 5. In this table the words Rx and POCUS should be specified near the respective total. 

Table 6: the % are missing.

Table 7: in this table I have not clear the number of fracture you considered to calculate SE and SP. PPV and NPV should also be shown

general observation

the generical diagnosis of fracture and the fracture types seem not to correspond in terms of diagnostic accuracy.

SE, SP, PPV, NPV  for each group of fracture type reported in the lines 5 and  6 of figure 1 should be shown.

C:I. for SE, SP, PPV, NPV should be reported.

In the abstract and in the discussion also PPV and NPV should be considered and discussed in evaluting the clinical performance of POCUS.

A question. You state that POCUS is an alternative to RX in the diagnosis of fracture, But you have false positives and negatives in the series. In view of those false cases do you believe that a clinically useful diagnosis of fracture can be made only by POCUS? In other words: POCUS is alternative or complementary to Rx.

Author Response

Round  2

Reviewer 2 Report

I have four more request.

1) you should explain why you believe that this paper is prospective (there is not a follow up evaluation)

2) I do not understand how the calculation of SE, SP, PPV and NPV was done.Could you show the contingency tables used to calculate those parameters?

3) line 142: the tests referred to "p" and "K" should be specified 

4) line 119: the state variable and the dependent variable should be specified.

Author Response

Medicina EISSN 1010-660X Published by MDPI AG, Basel, Switzerland RSS E-Mail Table of Contents Alert
Back to Top